# Practice of preventive measures and vaccine hesitance for COVID 19 among households in The Gambia, 2021: Study protocol

Bakary Sanneh[1]*, Sainey Sanneh[2], Sharmila Lareef-Jah[3], Buba Darboe[4], Lamin L. Dibba[5], Lamin F. Manjang[2], Yorro Bah[2], Jalimory Suso[2], Phebian Ina Grant Sagnia[2], Modou Njai[4], Sana M. Sambou[6]

1 National Public Health Laboratories, Ministry of Health, Bertil Herding Highway, Kotu, The Gambia, 2 Directorate of Health Research, Ministry of Health, Banjul, The Gambia, 3 World Health Organization, Country Office, Kotu, The Gambia, 4 Directorate of Health Promotion and Education, Ministry of Health Bertil Herding Highway, Kotu, The Gambia, 5 Gambia Bureau of Statistics, Ministry of Finance & Economic Affairs, Serekunda, The Gambia, 6 Epidemiology and Disease Control Unit, Ministry of Health Bertil Herding Highway, Kotu, The Gambia

☉ These authors contributed equally to this work.

* sheikbakary@yahoo.com

**Funding:** Will be Funded by World Health Country Office,The Gambia.Funding number of AFGMB2018552. The involvement of WHO country

## Abstract

The first imported confirmed case of COVID 19 was reported in The Gambia on 16th of March 2020 which led to the implementation of relevant public health interventions to prevent further importation and spread of the virus. However, by 8th November 2021, the country had registered cumulatively 9.980 COVID-19 confirmed infection and 341 deaths. The country has developed and implemented Risk Communication and Community Engagement (RCCE) Action Plan since the declaration by WHO that COVID-19 outbreak was a global public health threat and its subsequent proclamation that outbreak was a pandemic. Despite these efforts to sensitize the communities, some Gambians are in denial and/or misinformed of the existence of infection in the country. It is also evident that social distancing and other restrictions have not been adequately implemented by the citizenry. Less 14% of The Gambian population have been vaccinated, and there is evidence of gross vaccine hesitancy and disbelief. There is urgent need to investigate the knowledge, attitude and practices among Gambians about preventive practices especially regarding accepting vaccination to control COVID 19. The proposed study will enrol 1200 households from seven Local Government Areas (LGAs). The findings of this study will inform the messaging and health promotion activities that will be used to better inform the population to ensure compliance and practice of preventive approaches (e.g., use of mask, vaccination)necessary to reduce the negative impact of COVID 19 outbreak in The Gambia. This will thus quicken the recovery process and the return to new normal life.

office staff does not influence the study design and represent the office in the study.

**Competing interests:** NO authors have competing interests.

# Background

Coronavirus disease 2019 (COVID 19) is a respiratory tract infection caused by a newly emergent coronavirus, that was first recognized in Wuhan, China, in December 2019 [1]. Subsequently WHO published a declaration of a global outbreak which was later upgraded to a pandemic [2, 3]. Genetic sequencing of the virus suggested that it is a beta-coronavirus closely linked to the SARS virus [4] A meta-analysis study reported that people with mild, moderate to severe COVID 19 infection might manifest symptoms and signs such as: cough, sore throat, high temperature, diarrhoea, headache, muscle or joint pain, fatigue, and loss or disturbance of sense of smell and taste [5]. These researchers revealed that patients with symptoms such as cough and fever had 21% increased risk to turn COVID 19 positive. Furthermore manifestation of loss of sense of smell or taste substantially increased the likelihood of COVID 19 infection by 8% [5]. The discoveries of some asymptomatic participants in different studies complicates the fight against such a pandemic [5]. Another meta-analysis study has found a post COVID 19 pandemic vaccination era mortality prevalence of 15%, associated with hospital admissions [6]. This study established that patients with acute respiratory distress syndrome were eight times more likely to die as compare to those who did not have the syndrome [6]. This resulted to the development of anxiety and worries among two third of the world population as demonstrated in a study [7]. Countries had implemented different public health strategies to prevent and/or contain the spread of the outbreak. These involved national lockdowns, border closures, social distancing, mask wearing and quarantine. The implementation of these intervention has led to a change from global social norms to a new world order. People in different parts of the world embraced, complied or adjusted to these new changes differently. For instance compliance rates for Covid-19 control measures in Tonga and Egypt were 93% and 43%, respectively [7].

The rapid evolution of mutant variants of COVID 19 increase the potential for the spread and severity of the disease. This setbacks back the global response plan to curb the pandemic. Hence, the only hope to revive and re-establish normalcy in global lifestyle was the discovery, approval and vaccination of the global populace. COVID 19 vaccines were approved in late 2020 and early 2021 for public use in countries across the world and the success of this public intervention will depended not only on vaccine safety and effectiveness but coverage rate to achieve herd immunity [8]. COVID 19 vaccine hesitancy meta-analysis study revealed that vaccine acceptance rate is region and country specific [8]. High acceptance were reported in Ecuador (97.0%), Malaysia (94.3%), Indonesia (93.3%) and China (91.3%); moderate acceptance was reported in Jordan (28.4%), Italy (53.7), Russia (54.9%), Poland (56.3%), US (56.9%), and France (58.9%) and lowest acceptance was reported in Kuwait (23.6%), Jordan (28.4%), from the Democratic Republic of the Congo (27.7%) Low rates of COVID 19 vaccine acceptance were also reported in the Middle East, Russia, Africa and several European countries [9]. Studies that investigate the factors leading to vaccine hesitancy are particularly needed in the Middle East and North Africa, Sub-Saharan Africa, Eastern Europe, Central Asia, Middle and South America. Addressing the scope of COVID 19 vaccine hesitancy in various countries is recommended as an initial step for building trust in COVID 19 vaccination efforts [9].

In response to the rapid spread of the virus, the World Health Organization (WHO) declared COVID 19 as a Public Health Emergency of International Concern (PHEIC) on 30th January 2020, and a pandemic on 11th March 2020. The Gambia embarked on developing and implement the National Preparedness and Response Plan (Response plan) [10]. The Gambia registered its first case of COVID 19 on the 16th of March 2020 and thus the Public Health Emergency Operational Centre was operationalized to using the Incidence Management System to guide and coordinate the national response to the outbreak [11, 12]. The country adopted and implemented an incident management plan that involves several measures to

interrupt importation and transmission of the virus nationwide [13]. These include the closing of schools, suspension of public gathering, closure of all non-essential public places, spatial distancing, and respiratory etiquette, restriction on number of passengers allowed on public transport, Airport closure by 17 March 2020 as well as, mandatory quarantine of travellers, isolation and care for infected and suspected cases. These measures led to the discomfort of the populace and such victims [14]. Massive community engagements activities were conducted to raise awareness about COVID 19 prevention and control practices and the provision of masks, hand sanitizers and hand washing facilities. A toll-free helpline (1025) which was instituted during the 2014 Ebola outbreak was reactivated. The call centre was operated 24 hours and 7 days by Ministry of Health [11]. These have facilitated the citizenry to make inquiries on COVID-19, seek support and advice if they notice any signs and symptoms or report possible suspects or complaints regarding people defying control measures [11]. However, by July 2020 most of the COVID 19 restrictions were loosened and the Airport was opened but testing continued for both returning and departing travellers that use the Airport [11, 15]. These travellers accounted for 90% of the daily testing's of COVID 19 samples at the National Public Health Laboratories. The COVID 19 rapid test has been introduced and testing facilities scaled up nation-wide. As per the 399th national situation COVID 19 report, a total of 341 COVID 19 related death (Crude Case-Fatality Ratio, 3.4%), with Cumulative confirmed cases of 9,980 was registered. The country witnessed intermittent outbreaks of COVID 19 in the following period July to September 2020 then a mild second wave in January to April 2021 and severe wave which was mainly associated with the delta variant from July to September 2021 [11]. COVID 19 virus lineages A and B have been detected and associated with the cause of second wave of the national outbreak. Lineage B constitutes almost 98% of the total genomes sequenced, with the sub-lineage B.1 being the most prevalent [16] whilst the third COVID 19 outbreak wave is confirmed to be mainly associated with the transmission of the delta variant [17]. Nationally, a study had phylogenetically confirmed two reinfections among healthy individuals, with a time lag of 5 months and 6 months, respectively [18]. This necessitated the introduction of vaccination to increase the attainment of herd immunity and thus the introduction of the public health intervention on 15th March 2021 [11]. As of 8th November the national COVID 19 coverage of targeted population with completed vaccination dose was 14.2% and whilst coverage of at least one dose was 15.1%. COVID 19 vaccination has the least coverage among all the vaccines ever administered in the country. This could be associated with misinformation and lack of belief in the existence of the outbreak and other associated factors [11]. There is limited information to explain why people are hesitant to take the COVID 19 vaccine despite massive community engagement activities being carried out. A lot of misinformation and conspiracy theories about the vaccine are spread through social platforms such as WhatsApp groups, Facebook and other electronic channels. There is an urgent need to understand the general public's awareness and perceptions on COVID 19. This is particularly important as the adherence to the control measures by the public has been viewed to be suboptimal. The information from this survey will not only provide data for further assessments but will help to develop targeted strategies to rapidly improve current behavioural and risk communication interventions. The objective of this study is to establish public knowledge, attitude and practice towards COVID 19 infection and its preventive measures; and to describe the factors associated with COVID 19 vaccine hesitancy in The Gambia.

## Hypothesis

High rate of misbelief of COVID 19 infection and effectiveness of its vaccine among household members have resulted to low vaccination coverage and adherence to preventive measure.

## The research questions are:

1. What is the level of knowledge about COVID 19 prevention, control and associated vaccination among household members?

2. What is the attitude of household members towards COVID 19 prevention, control and associated vaccinations?

3. What are the practices of household members towards COVID 19 prevention, control and associated vaccinations?

## Methodology

### Study design

A cross-sectional study design will be employed to explore the knowledge, attitude and practices of households in the context of COVID 19 prevention and control practices which includes the willingness to accept vaccination.

### Study sites

Enumerated areas from the seven LGAs in the country will be randomly selected. These LGAs will be further stratified as rural, urban, peri-urban areas to determine the disparity of understanding and practices of prevention and control measures of COVID 19 in the country amidst access to social amenities.

## Study population

### Eligibility criteria

*Inclusion criteria*:

- Individuals identified for recruitment into the selected Enumeration Areas (EAs) and households in particular.

*Exclusion criteria*:

- Participants less than 18 years of age.

## Sampling methodology

### Sample design and selection

**Sample size.** Considering that the true variability of the characteristic of interest in the population is unknown in advance, sample size in this survey was computed taking into account the total number of households in the country, the sample design and method of estimation, and the response rate. Due to lack of a key indicator (P) to be measured by the survey, a value to 50% is assumed to give the maximum level of variability. Given that it's satisfactory if the true population proportion is within ±5, an anticipated response rate of 90% was used to effectively achieve the desired precision for the estimates. Thus, the sample size for the survey is computed as follows:

$$n = \frac{deft^2 \times (1/P - 1)}{\alpha^2}$$

For the purpose of the Survey, the following assumptions inform the sample size calculation:

- P = 50% or 0.5

- Sample size and design effect = 1.5

- Level of Confidence = 95% i.e. $\alpha$ = 0.05

- Margin of Error = ± 5%

- Response rate = 90% or 0.90

Using the formula and the parameters above, a minimum sample size of 900 was computed at the national level. This sample size will give a coefficient of variation (CV) which is also known as relative standard error (RSE) of 5 per cent at the national level. Adjusting the sample using a response rate of 90 per cent will give 1,000 household interviews. Given that the country is administratively divided into eight LGAs, a multi-stage cluster sampling will be used to select samples in three stages from the frame. In the first stage, after sorting the frame by LGA and Region (i.e. Urban and Rural), 100 EAs will be independently selected using probability proportional to household size. Table 1 below shows a summary of the sample design.

## Sample frame

The list containing Enumeration Areas (EAs) of all the eight geographic and administrative regions in The Gambia and their respective households and population obtained from 2013 Gambia Population and Housing Census Frame will be used as the sampling frame as shown in Fig 1. Consequently, with the data and cartographic information, all the EAs will be selected at the first stage based on the design proposed for this study.

**Sample design.** For this study, a multi-stage stratified cluster sample design will be used to select the eligible respondents in three stages. In the first stage, after sorting the frame, EAs will be selected using Probability Proportional to the Size (PPS) of EA. As defined in the 2013 Population and Housing Census, EA size is the number of residential households residing in the EA during the population census.

**Selection scheme.** In the second stage, EAs will be sorted and stand as strata in the new frame. In each of the respective selected EAs, 10 households will be selected using equal probability systematic sampling procedures. Before the data collection, a household listing operation for each of the selected EAs will be carried out by field supervisors and enumerators. All the households will be listed and ten (10) households will be selected using simple random

**Table 1. Allocation of Enumeration Areas (EAs) to different strata.**

| LGA | Total EAs | Total Urban | Total Rural | Total Selected EAs | Selected Urban EAs | Selected Rural EAs |
|---|---|---|---|---|---|---|
| Banjul | 74 | 74 | - | 2 | 2 | - |
| Kanifing | 773 | 773 | - | 22 | 22 | - |
| Brikama | 1,466 | 1,338 | 128 | 45 | 23 | 22 |
| Kerewan | 493 | 106 | 387 | 4 | 2 | 2 |
| Mansakonko | 204 | 32 | 172 | 8 | 5 | 3 |
| Kuntaur | 237 | 16 | 221 | 5 | 2 | 3 |
| Janjanbureh | 297 | 43 | 254 | 6 | 3 | 3 |
| Basse | 554 | 158 | 396 | 8 | 2 | 6 |
| **Total** | **4,098** | **2,540** | **1,558** | **100** | **61** | **39** |

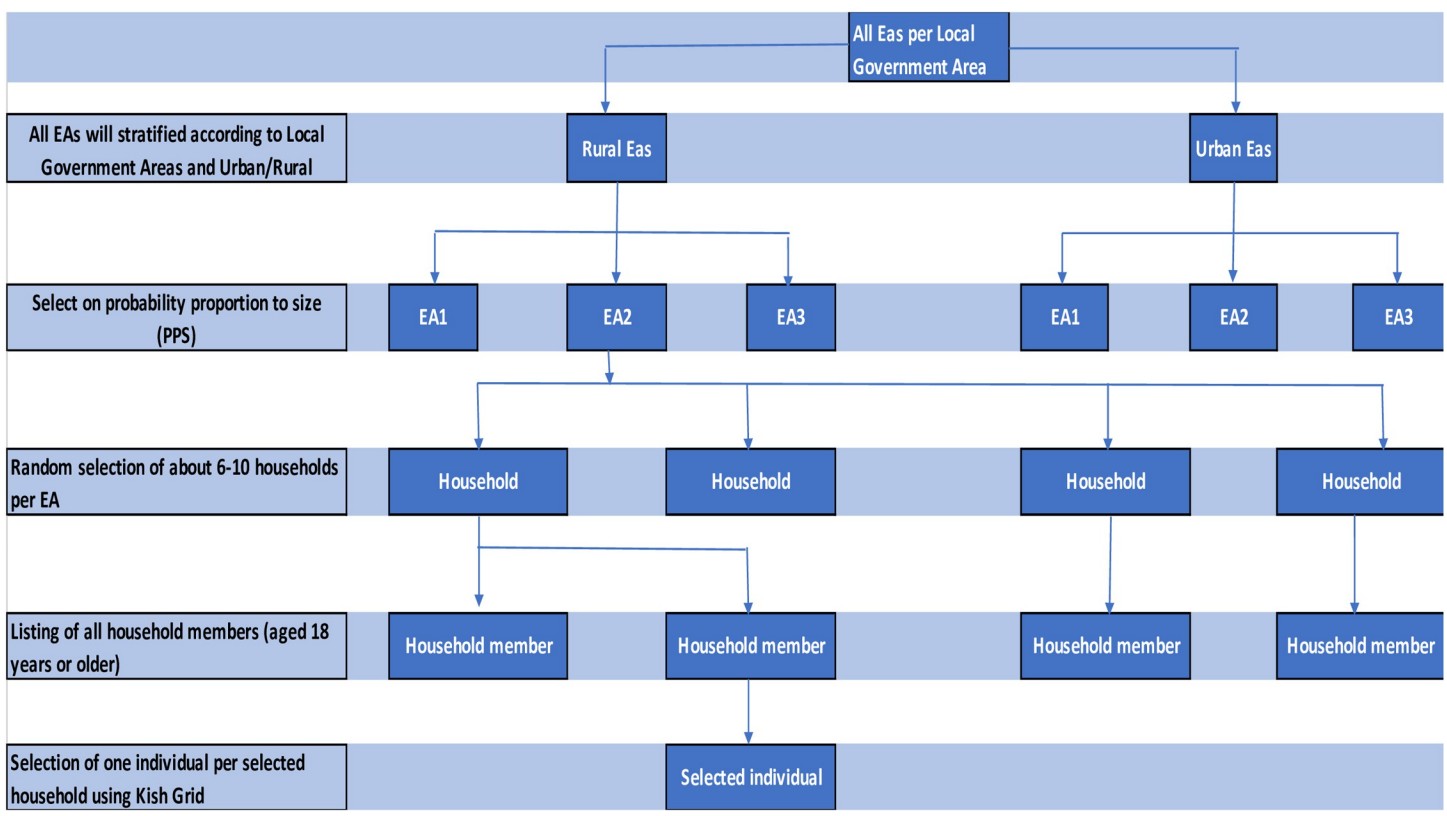

**Fig 1. Sampling framework.**

sampling (third stage). The identified households to be interviewed will be visited by the field staff. Strictly, it is important to note that survey interviewer must interview only the selected households. To prevent bias, no replacements and no changes of the selected households will be permitted in the implementing stages. One individual aged 18 years and above will be selected randomly for the interview using knish grid.

Table 1 below gives the allocation of selected EAs in the various domains. The population of those 18 years and above from the 2018 Labour Force Survey was used in allocating the sample to the urban and rural strata in each LGA in order to ensure that the sample is representative of the target group.

Distribution of selected households for various LGAs is shown in Table 2.

**Listing procedures.**   All 100 selected EAs will be listed in order to update the number of households in each frame as in Fig 1. The currently ongoing Integrated Household Survey 2020/2021 listing template will be used to obtain the total eligible households per cluster. This is a very crucial step as the computation of the second-stage probability of selection will depend entirely on the accuracy and reliability of these numbers. It is imperative to explore digital data collection this time around to enable the coordinating team ensure quality as well as select unbiased second-stage samples at the central level. Also, the coordinates of the selected structures can easily be drawn on the map for easy identification thereby saving the enumerators from identifying the wrong structures selected to participate in the survey.

**Weighting procedures.**   By design, complex sampling involving selection at different stages will be used in the study. Inclusion probabilities will be calculated from each stage of selection and as a result, sampling weights will be computed and used for the analysis to ensure

**Table 2. Distribution of selected households per local government areas.**

| LGA | Total Households | Selected Households |
|---|---|---|
| Banjul | 7272 | 20 |
| Kanifing | 69890 | 220 |
| Brikama | 103664 | 450 |
| Kerewan | 11965 | 40 |
| Mansakonko | 27862 | 80 |
| Kuntaur | 10957 | 50 |
| Janjanbureh | 14451 | 60 |
| Basse | 34641 | 80 |
| **Total** | **280,702** | **1,000** |

representativeness of the sample. The selection probabilities at each stage will be documented and the inverse of these probabilities is the basic weight also known as the design weight.

The household design weight is the inverse of the overall selection probability of the household which is computed as the product of the first stage (PSU/cluster) selection probability and the second stage (household) selection probability.

The number of eligible individuals per household multiplied by the corresponding household design weight in the cluster gives the individual weights.

Survey Instrument and Variables.

A questionnaire was adapted to The Gambia context from the standardized IFRC, UNCIF and WHO Risk Communication and Community Engagement (RCCE) Action Plan [19] will be used for the survey(provided as supporting documents). Although it has been adapted to the local context, the questionnaire will be pretested and findings will be used to further refine the questions. The survey items are organised in four sections namely demographics, knowledge, attitudes and practices and consists of primarily close-ended questions. Consent will be obtained before administration of the questionnaire.

**Data management, analysis and presentation.** Data collection will be done by trained and experienced enumerators. A virtual meeting with enumerators will be held daily to review the process and identify challenges and successes. An exploratory data analysis (EDA) will be conducted after the collection of the data in order to make a quick check into the consistency and validity of the data. Subsequently, necessary data cleaning will be done to prepare the data for analysis. Analysis of the data collected will be conducted using the International Business Machines Statistical Package for Social Sciences (IBM SPSS). Frequencies will be run to explore missing responses and out-of-range values for each of the demographic variables as well as those used for the main analysis. The distribution of the data will be subjected to normality test using the Shapiro-Wilk test of normality. As the data collection is premised on assessing the knowledge, attitude and practices towards COVID 19 and willingness to accept vaccination, analysis will focus on key demographic variables such as gender, location and age. Cross tabulations with independent variables and KAP of COVID 19 prevention will be done to obtain answer the set study objectives. The analysis of the data will focus on the three components of the survey namely knowledge, attitude and practice and association with willingness to accept COVID 19 vaccinations. Knowledge assesses the amount of knowledge respondents have about COVID 19 and the source(s) of information and associated vaccination belief using chi square at p value of less than or equal to 0.05. Attitude questions will address people's attitude towards the pandemic such as their belief that it is a reality while practice questions will focus on the activities people are engaged in in preventing themselves,

their families and their neighbours from the disease through vaccination and other practices. Key results will be presented in the form of tables and graphs and the necessary statistical interpretations will be done in the form of an analytical report for policy makers. The study finding will be presented stakeholders, partners in COVID 19 response.

## Potential risks

There is anticipated risk of infection to the public or data collectors given that we are in an active pandemic. Therefore, to minimise such risk all data collectors and drivers will be trained on basic IPC procedures, must be vaccinated. In addition, interviews will be conducted maintaining the WHO-recommended social distancing guidelines, and compulsory wearing of face mask.

## Ethical consideration

The study has been approved The Gambia Government/MRCG Laboratories Joint Ethics Committee (Project ID/ethics ref: 22699) and from the Ministry of Health for the study protocol and procedures of informed consent. Written informed consent will be obtained from all participants. The participants will be provided with adequate information about purpose of the study in a language they best understand. Information from participants will be treated confidentially and will not be shared with anyone except study team members.

## Expected outcome and implication of findings

This survey will reveal an accurate picture of what the general population knows about the pandemic and their attitudes and practices towards prevention and control measures for this disease and their willingness to accept vaccination. It will assess the huge investment the government and partners and donors had facilitate the implementation of the Communication and Community Engagement (RCCE) Action Plan. This will help to guide the review and updating RCCE plan to impact better behavioural change toward eventually positive acceptance and practices prevention and control of COVID 19 and subsequent increase for vaccination coverage which will assure the attainment of herd immunity. The finding of this study will be published in peer open journals and the dataset will be freely accessible and shared with the publishing journals.

## Supporting information

**S1 File. Covid19 kap information sheet & consent form.**
(DOCX)

**S2 File. COVID 19 kap survey tool questionnaires.**
(DOCX)

## Acknowledgments

The authors appreciated the review and inputs of the Gambia Government and MRC Joint Ethics Committee.We are grateful toward Mr Ebrima Joof, a PhD student at University of Nottingham, United Kingdom and Dr Dave Mill for the proofreading the study protocol.

## Author Contributions

**Conceptualization:** Bakary Sanneh, Sainey Sanneh, Sharmila Lareef-Jah, Buba Darboe, Modou Njai, Sana M. Sambou.

**Funding acquisition:** Sainey Sanneh.

**Methodology:** Bakary Sanneh, Sainey Sanneh, Sharmila Lareef-Jah, Buba Darboe, Lamin L. Dibba, Lamin F. Manjang, Yorro Bah, Jalimory Suso, Phebian Ina Grant Sagnia, Modou Njai.

**Project administration:** Sainey Sanneh, Sharmila Lareef-Jah, Buba Darboe.

**Software:** Lamin L. Dibba.

**Supervision:** Sainey Sanneh, Sharmila Lareef-Jah, Buba Darboe, Modou Njai.

**Validation:** Bakary Sanneh, Sainey Sanneh, Sharmila Lareef-Jah, Lamin L. Dibba, Yorro Bah, Sana M. Sambou.

**Visualization:** Sainey Sanneh, Lamin L. Dibba, Yorro Bah, Jalimory Suso, Phebian Ina Grant Sagnia, Sana M. Sambou.

**Writing – original draft:** Bakary Sanneh, Sainey Sanneh, Sharmila Lareef-Jah, Buba Darboe, Lamin L. Dibba, Lamin F. Manjang, Phebian Ina Grant Sagnia, Sana M. Sambou.

**Writing – review & editing:** Bakary Sanneh, Lamin F. Manjang, Yorro Bah, Jalimory Suso, Phebian Ina Grant Sagnia, Modou Njai, Sana M. Sambou.

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
