## [Decision Letter · Decision Letter 0]

29 Nov 2021

PONE-D-21-36320Knowledge, Attitude and Practices of COVID19 and associated vaccine hesitance among Households in the Gambia, 2021.Study protocol.PLOS ONE

Dear Dr. Sanneh,

Thank you for submitting your manuscript to PLOS ONE. After careful consideration, we feel that it has merit but does not fully meet PLOS ONE’s publication criteria as it currently stands. Therefore, we invite you to submit a revised version of the manuscript that addresses the points raised during the review process.

We look forward to receiving your revised manuscript.

Kind regards,

Sheikh Mohd Saleem, MBBS, MD

Academic Editor

PLOS ONE

Journal Requirements:

" ext-link-type="uri" xlink:type="simple">https://journals.plos.org/plosone/s/file?id=ba62/PLOSOne_formatting_sample_title_authors_affiliations.pdf"

The study protocol will be funded by WHO, The Gambia Country.

The study protocol will be funded by WHO, The Gambia Country.

3. We noted in your submission details that a portion of your manuscript may have been presented or published elsewhere. "Previous Interactions

Indicate whether you have had any of the following previous interactions about this manuscript (check all that apply):

I have had previous interactions about this manuscript with an Academic Editor of this journal.

↳ Describe your previous interactions regarding this manuscript, and include the name of the editor you interacted with.

The manuscripts were published and others declined" Please clarify whether this [conference proceeding or publication] was peer-reviewed and formally published. If this work was previously peer-reviewed and published, in the cover letter please provide the reason that this work does not constitute dual publication and should be included in the current manuscript.

Additional Editor Comments:

Thankyou for submitting your manuscript to PLOSONE. As per the review comments, Minor revision is suggested.

Reviewers' comments:

Reviewer's Responses to Questions

**Comments to the Author**

1. Does the manuscript provide a valid rationale for the proposed study, with clearly identified and justified research questions?

Reviewer #1: Yes

Reviewer #2: Partly

Reviewer #3: Yes

2. Is the protocol technically sound and planned in a manner that will lead to a meaningful outcome and allow testing the stated hypotheses?

Reviewer #1: Yes

Reviewer #2: Partly

Reviewer #3: Yes

3. Is the methodology feasible and described in sufficient detail to allow the work to be replicable?

Reviewer #1: Yes

Reviewer #2: Yes

Reviewer #3: Yes

4. Have the authors described where all data underlying the findings will be made available when the study is complete?

Reviewer #1: Yes

Reviewer #2: No

Reviewer #3: No

5. Is the manuscript presented in an intelligible fashion and written in standard English?

Reviewer #1: Yes

Reviewer #2: Yes

Reviewer #3: Yes

6. Review Comments to the Author

You may also provide optional suggestions and comments to authors that they might find helpful in planning their study.

Reviewer #1: This is a KAP study protocol. The protocol is presented in a manner that it can be replicated by any scholar anywhere and anytime; is scientifically sound and it can shape the area of KAP surveys in future.

This protocol should be given chance

Reviewer #2: Topic selection of this research is very relevant to this current senario but, objectives of this study is not well defined. Research hypothesis is well expalined. Methodology part is well described. Regarding the analysis part, how to show the association between knowledge, attitude and practice is not explained properly.

Overall content is nice with some spelling mistakes and some gramatical errors. There is also some errors in citation part of article.

Reviewer #3: This is pertinent work. The followings are minor comments

Abstract: There is a need to focus more on the research protocol

Background: Will benefit of drafting a chart with trend of cases, policies, vaccination etc.

Research questions are rather generic

Supporting documents do not include questionnaire of informed consent

Utilization of the findings can be clearer

7. PLOS authors have the option to publish the peer review history of their article (what does this mean?). If published, this will include your full peer review and any attached files.

Reviewer #1: **Yes: **Abdulrahman Ahmad

Reviewer #2: No

Reviewer #3: No

---

## [Author Response · Author response to Decision Letter 0]

30 Nov 2021

Letter responding to the reviewers comments has been uploaded.

---

## [Decision Letter · Decision Letter 1]

20 Jan 2022

PONE-D-21-36320R1Knowledge, Attitude and Practices of COVID19 and associated vaccine hesitance among Households in the Gambia, 2021.Study protocol.PLOS ONE

Dear Dr. %Sanneh%,

Thank you for submitting your manuscript to PLOS ONE. After careful consideration, we feel that it has merit but does not fully meet PLOS ONE’s publication criteria as it currently stands. Therefore, we invite you to submit a revised version of the manuscript that addresses the points raised during the review process.

ACADEMIC EDITOR: Based on the reviewers comments and merits of your manuscript, Kindly revise your manuscript based on the commenets given at the earliest.==============================

We look forward to receiving your revised manuscript.

Kind regards,

Sheikh Mohd Saleem, MBBS, MD

Academic Editor

PLOS ONE

Journal Requirements:

Reviewers' comments:

Reviewer's Responses to Questions

**Comments to the Author**

1. Does the manuscript provide a valid rationale for the proposed study, with clearly identified and justified research questions?

Reviewer #2: Partly

2. Is the protocol technically sound and planned in a manner that will lead to a meaningful outcome and allow testing the stated hypotheses?

Reviewer #2: Partly

3. Is the methodology feasible and described in sufficient detail to allow the work to be replicable?

Reviewer #2: Yes

4. Have the authors described where all data underlying the findings will be made available when the study is complete?

Reviewer #2: No

5. Is the manuscript presented in an intelligible fashion and written in standard English?

Reviewer #2: Yes

6. Review Comments to the Author

You may also provide optional suggestions and comments to authors that they might find helpful in planning their study.

Reviewer #2: Topic selection of this research is very relevant to this current senario but, objectives of this study is not well defined. Research hypothesis is well expalined. Methodology part is well described. Regarding the analysis part, how to show the association between knowledge, attitude and practice is not explained properly.

Overall content is nice with some spelling mistakes and some gramatical errors. There is also some errors in citation part of article.

7. PLOS authors have the option to publish the peer review history of their article (what does this mean?). If published, this will include your full peer review and any attached files.

Reviewer #2: No

---

## [Decision Letter · Decision Letter 2]

11 Mar 2022

PONE-D-21-36320R2Knowledge, Attitude and Practices of COVID19 and associated vaccine hesitance among Households in the Gambia, 2021.Study protocol.PLOS ONE

Dear Dr. Sanneh,

Thank you for submitting your manuscript to PLOS ONE. After careful consideration, we feel that it has merit but does not fully meet PLOS ONE’s publication criteria as it currently stands. Therefore, we invite you to submit a revised version of the manuscript that addresses the points raised during the review process.

If applicable, we recommend that you deposit your laboratory protocols in protocols.io to enhance the reproducibility of your results. Protocols.io assigns your protocol its own identifier (DOI) so that it can be cited independently in the future. For instructions see: https://journals.plos.org/plosone/s/submission-guidelines#loc-laboratory-protocols. Additionally, PLOS ONE offers an option for publishing peer-reviewed Lab Protocol articles, which describe protocols hosted on protocols.io. Read more information on sharing protocols at https://plos.org/protocols?utm_medium=editorial-emailutm_source=authorlettersutm_campaign=protocols.

We look forward to receiving your revised manuscript.

Kind regards,

Sheikh Mohd Saleem, MBBS, MD

Academic Editor

PLOS ONE

Reviewers' comments:

Reviewer's Responses to Questions

**Comments to the Author**

1. Does the manuscript provide a valid rationale for the proposed study, with clearly identified and justified research questions?

Reviewer #2: Yes

Reviewer #4: Yes

2. Is the protocol technically sound and planned in a manner that will lead to a meaningful outcome and allow testing the stated hypotheses?

Reviewer #2: Partly

Reviewer #4: Yes

3. Is the methodology feasible and described in sufficient detail to allow the work to be replicable?

Reviewer #2: Yes

Reviewer #4: Yes

4. Have the authors described where all data underlying the findings will be made available when the study is complete?

Reviewer #2: Yes

Reviewer #4: Yes

5. Is the manuscript presented in an intelligible fashion and written in standard English?

Reviewer #2: Yes

Reviewer #4: No

6. Review Comments to the Author

You may also provide optional suggestions and comments to authors that they might find helpful in planning their study.

Reviewer #2: Topic selections and plan of execution of this research well explained. This study will definetly help to find the KAP regarding COVID-19 among the people. Comments has been well addressed.

Reviewer #4: (Corrections below refer to the WORD version clean copy of the covid 19 manuscript _ Revised 2022.docx accessed via click)

Summary of the research and overall impression

The topic of this study protocol is of public health significance at the moment .The investigation into people’s practice of COVID 19 prevention and control measures and factors associated with vaccine hesitance is not only interesting but the results will be useful in the Gambia and other sub-Saharan African settings with similar observed resistance to these best practices. The manuscript is presented in an intelligible fashion but the english could be improved upon to bring it up to standard. There are quite a number of grammatical errors in the Abstract and Background sections. The manuscript would benefit from grammar editing. If the following major revisions are carried out, I strongly recommend it for publication in PLOS ONE.

Major Revision

Title: A preferred title (or a similar one along these lines) would be “Knowledge, Attitude towards COVID19 and its prevention; practice of preventive measures; and COVID 19 vaccine hesitance among Households in the Gambia, 2021: Study protocol.

The problems with the title as written by the authors are:

- “practices of COVID 19”. It’s the preventive measures of Covid 19 that are practiced and not COVID 19.

- Knowledge and attitude towards COVID 19 prevention is not clearly brought out in the title. However, one may assume it is subsumed in just COVID 19.

Abstract: Would benefit from grammar editing.

The following statement should include the other issue being investigated i.e vaccine hesitance “There is urgent need to investigate the knowledge, attitude and practices among the Gambians about preventive practices to control COVID-19”.

Background:

Would benefit from grammar editing.

The following sentences should be rewritten for clarity, their meanings are not clear.

- “It was alluded that some of the study participants were positive for COVID 19 different studies had no signs and symptoms of COVID 19 and thus blurring the fight against such a pandemic[5].”

- “In the post vaccination era of COVID 19 pandemic the pooled prevalence of 15% mortality with found to the associated in hospital admissions [6].

Could the authors provide a reference for this statement from previous studies if available “This is particularly important as the adherence to the control measures by the public has been viewed to be suboptimal”.

The authors state “ There is limited information to explain why people are hesitant to take the COVID 19 vaccinations despite massive community engagement activities done”. While this is true, the authors are advised to cite what little information is available after a thorough search of recent literature”.

The last sentence which states the objectives of the study should be changed to “The objective of this study is to establish the knowledge, attitude towards COVID 19 infection and its prevention; practice of preventive measures; and describe the factors associated with Covid 19 vaccination hesitance in the Gambia”.

Hypothesis:

Could the hypothesis be rephrased in a more scientific in language

Research questions:

The authors should add “prevention and control” after “COVID 19” in the three Research questions. Also a fourth research question is needed to cover “the factors associated with Covid 19 vaccination hesitance” mentioned in the objective of the study.

Sampling Methodology:

The title of Table 2 was same title of Table 1. This must be an error. The correct title would appear to be the sentence above the table i.e “Distribution of selected households for various LGAs”

Data Management, Analysis and Presentation:

The data analysis should include an analysis of factors associated with vaccine hesitance as stated in the objectives: suggested are cross tabulations with independent variables and KAP of COVID 19 prevention

7. PLOS authors have the option to publish the peer review history of their article (what does this mean?). If published, this will include your full peer review and any attached files.

Reviewer #2: No

Reviewer #4: No

---

## [Author Response · Author response to Decision Letter 2]

27 Mar 2022

The response to varied comments and concerns have been addressed in the uploaded letter to the reviewer.

---

## [Editor Report · Decision Letter 3]

7 Apr 2022

PONE-D-21-36320R3: Practice of preventive measures and vaccine hesitance for COVID 19 among Households in The Gambia, 2021: Study protocol.PLOS ONE

Dear Dr. Sanneh,

Thank you for submitting your manuscript to PLOS ONE. After careful consideration, we feel that it has merit but does not fully meet PLOS ONE’s publication criteria as it currently stands. Therefore, we invite you to submit a revised version of the manuscript that addresses the points raised during the review process.

If applicable, we recommend that you deposit your laboratory protocols in protocols.io to enhance the reproducibility of your results. Protocols.io assigns your protocol its own identifier (DOI) so that it can be cited independently in the future. For instructions see: https://journals.plos.org/plosone/s/submission-guidelines#loc-laboratory-protocols. Additionally, PLOS ONE offers an option for publishing peer-reviewed Lab Protocol articles, which describe protocols hosted on protocols.io. Read more information on sharing protocols at https://plos.org/protocols?utm_medium=editorial-emailutm_source=authorlettersutm_campaign=protocols.

We look forward to receiving your revised manuscript.

Kind regards,

Sheikh Mohd Saleem, MBBS, MD

Academic Editor

PLOS ONE

Additional Editor Comments:

Kindly address the comments provided by the reviewers
---

## [Author Response · Author response to Decision Letter 3]

22 Apr 2022

We have reviewed comments of the reviewer and all comments were addressed in the latest version of the submission

---

## [Decision Letter · Decision Letter 4]

8 Jun 2022

: Practice of preventive measures and vaccine hesitance for COVID 19 among Households in The Gambia, 2021: Study protocol.

PONE-D-21-36320R4

Dear Dr. Sanneh,

We’re pleased to inform you that your manuscript has been judged scientifically suitable for publication and will be formally accepted for publication once it meets all outstanding technical requirements.

Kind regards,

Sheikh Mohd Saleem, MBBS, MD

Academic Editor

PLOS ONE

Additional Editor Comments (optional):

Reviewers' comments:

Reviewer's Responses to Questions

**Comments to the Author**

1. Does the manuscript provide a valid rationale for the proposed study, with clearly identified and justified research questions?

Reviewer #4: Yes

2. Is the protocol technically sound and planned in a manner that will lead to a meaningful outcome and allow testing the stated hypotheses?

Reviewer #4: Yes

3. Is the methodology feasible and described in sufficient detail to allow the work to be replicable?

Reviewer #4: Yes

4. Have the authors described where all data underlying the findings will be made available when the study is complete?

Reviewer #4: Yes

5. Is the manuscript presented in an intelligible fashion and written in standard English?

Reviewer #4: Yes

6. Review Comments to the Author

You may also provide optional suggestions and comments to authors that they might find helpful in planning their study.

Reviewer #4: The issues of concern raised in the previous review have been addressed. However, references for the following statements in the "Background" have still not been provided: “This is particularly important as the adherence to the control measures by the public has been viewed to be suboptimal” .“There is limited information to explain why people are hesitant to take the COVID 19 vaccinations despite massive community engagement activities done”. If no references could be found , I suggest the authors state this and if it is anecdotal evidence, state as much. Also minor grammar editing is still needed, for example, the word "to" should be inserted between "presented" and "stakeholders" in the last line in the "Data Management, Analysis and Presentation" section.

7. PLOS authors have the option to publish the peer review history of their article (what does this mean?). If published, this will include your full peer review and any attached files.

Reviewer #4: **Yes: **Dr Adaoha Pearl AGU

---

## [Editor Report · Acceptance letter]

10 Aug 2022

PONE-D-21-36320R4 

Practice of preventive measures and vaccine hesitance for COVID 19 among Households in The Gambia, 2021: Study protocol. 

Dear Dr. Sanneh:

I'm pleased to inform you that your manuscript has been deemed suitable for publication in PLOS ONE. Congratulations! Your manuscript is now with our production department. 

Kind regards, 

on behalf of

Dr. Sheikh Mohd Saleem 

Academic Editor

PLOS ONE